

# Expression level and clinical significance of LncRNA PVT1 in the serum of patients with LEASO

Xiaoxue Su[1,*], Xiaoqing Yuan[2,*], Fenghui Li[3], Guinan Yang[1], Liunianbo Du[4], Fule Zhao[1], Rui Zhao[1], Minghui Ou[5]

[1] Qingdao University, Qingdao, China
[2] Guangzhou Kingmed Diagnostics Group, Guangzhou, China
[3] Qingdao Municipal Hospital Group, Qingdao, Shandong, China
[4] Dalian Medical University, Dalian, China
[5] Department of Vascular Surgery, Qingdao Municipal Hospital Group, Qingdao, China
[*] These authors contributed equally to this work.

Corresponding author
Minghui Ou, oumhmed@sina.cn

## ABSTRACT

**Objective**. Our study aims to investigate the long non-coding RNA plasmacytoma variant translocation 1 (lncRNA PVT1) in lower extremity arteriosclerosis obliterans (LEASO) patient serum and its clinical significance in LEASO.

**Patients and Methods**. From July 2021 to April 2022, 133 LEASO patients diagnosed at the Qingdao Municipal Hospital were included. Among them, 44 complicated with coronary artery disease (CAD) were classified as the LEASO with CAD group. The remaining 89 were marked as the LEASO group, which was classified into single ($n = 48$) and double ($n = 41$) lower limb groups, with the former being subclassified into the left ($n = 28$) and right ($n = 20$) lower limb groups based on the affected sites. Fifty healthy individuals who came to our hospital for physical examination during the same period were randomly included and defined as the Healthy Control group. PVT1 expression was detected in serum samples from each group using a quantitative reverse transcriptase-polymerase chain reaction , and differences in expression levels were calculated. The ankle-brachial index (ABI) of patients in the LEASO group was measured using a sphygmomanometer, and its correlation with PVT1 was analyzed. Clinical data and laboratory test results (including blood routine, liver and renal function, and blood lipids) were collected for all patients upon admission. Logistic regression analyses were performed to determine the influence of PVT1 and laboratory test results on LEASO. The diagnosis and prediction of LEASO were obtained by combing PVT1 with laboratory test indicators.

**Results**. It was found that lncRNA PVT1 expression was the highest in the serum of the LEASO with CAD group, followed by the LEASO and control groups ($P < 0.05$). Within the LEASO group, no significant difference in PVT1 expression was seen between the left and right limbs ($P > 0.05$), nor between the single and double lower limb groups. Furthermore, the PVT1 expression increased with the Rutherford grades, indicating a negative correlation between PVT1 and ABI. Logistic regression analysis revealed that triglycerides (OR = 2.972, 95% CI [1.159–7.618]), cholesterol (OR = 6.655, 95% CI [1.490–29.723]), C-reactive protein (OR = 1.686, 95% CI [1.218–2.335]), and PVT1 (OR = 2.885, 95% CI [1.350–6.167]) were independent risk factors for LEASO. Finally, strong sensitivity was observed in the receiver operating characteristic curve

when combining PVT1 with meaningful laboratory indicators to diagnose and predict LEASO.

**Conclusion**. lncRNA PVT1 promotes LEASO occurrence and progression and is related to atherosclerosis severity. The expression of PVT1 was negatively correlated with ABI. Logistic regression analysis suggested that blood lipid levels and inflammatory reactions might be related to LEASO occurrence. PVT1 was incorporated into laboratory indicators to predict LEASO. The subject's working curve area was large, and the prediction results were highly sensitive.

# INTRODUCTION

Lower extremity arteriosclerosis obliterans (LEASO) is an arterial disease that involves atherosclerosis (AS) in lower extremity arteries. LEASO thickens and hardens the intima of lower limb arteries, inducing atherosclerotic plaque formation and calcification. These changes may be accompanied by thrombosis. As a result, the arterial lumen becomes narrowed or even occluded, leading to limb ischemia. Clinical manifestations can be detected on the affected limbs, such as cold sensation, numbness, pain, intermittent claudication and ulceration or necrosis of the toe or foot (*Jakic et al., 2019*).

LEASO is common in patients aged 55–75 years (*Malgor et al., 2015*). High-risk factors for the disease include hyperlipidemia, hypertension, smoking, diabetes and obesity. With the aging of the Chinese population and the changes in diet structure, the incidence rates of LEASO continue to increase (*Fowkes et al., 2013*). Critical limb ischemia (CLI) is the end-stage symptom of LEASO. CLI has high mortality rates, particularly among patients with cardiovascular events. The reported mortality rate of CLI patients with a 6-month post-diagnosis period is as high as 20%, and that after diagnosis of 5 years is more than 50% (*Teraa et al., 2016*). In addition to poor survival, limb-salvage prognosis in CLI patients is unfavorable, with 6-month major amputation rates ranging from 10% to 40% (*Abu Dabrh et al., 2015*; *Becker et al., 2011*; *Norgren et al., 2007*). The early stages of LEASO are usually characterized by no obvious clinical symptoms. The lack of specific and sensitive LEASO markers makes diagnosis and treatment difficult in the early stages of the disease. Most patients have already presented severe symptoms when they saw a doctor, resulting in CLI. In this sense, identifying serum markers is essential for LEASO detection. Several studies have highlighted the importance of long non-coding RNAs (lncRNAs) in endothelial dysfunction, vascular smooth muscle cell phenotypic transformation, reverse cholesterol transport and foam cell formation (*Zhou et al., 2016*). This research proposes that lncRNAs may contribute to AS development.

LncRNA plasmacytoma variant translocation 1 (PVT1) was initially identified as a common retroviral integration site in mice with leukemia (*Webb, Adams & Cory, 1984-1985*). LncRNA PVT1 locates on chromosome 8q24 and is 1,716 nucleotides long. It is

involved in the pathogenesis of many tumors (*Colombo et al., 2015*). *Quan et al. (2020)* found that patients with coronary artery disease (CAD) had elevated levels of lncRNA PVT1. Moreover, the expression level of PVT1 was positively correlated with the Gensini score, indicating that PVT1 is involved in atherogenesis (*Quan et al., 2020*). Whether LncRNA PVT1 is implicated in the progression of LEASO has not yet been demonstrated.

In LEASO screening and diagnosis, the ankle-brachial index (ABI) is widely used (*Gerhard-Herman et al., 2017*; *Aboyans et al., 2018*). On the other hand, ABI has poor sensitivity when diagnosing LEASO in patients with diabetes and chronic kidney disease due to medial calcification in the below-the-knee arteries. In these patients, the measured ABI values are often inaccurate despite the presence of peripheral arterial disease in the lower extremities. In this context, relying solely on ABI values may lead to missed diagnoses of LEASO (*Maruhashi et al., 2021*; *Emanuele, Buchanan & Abraira, 1981*).

In this study, lncRNA PVT1 expression in LEASO patient serum was examined to investigate its role in LEASO progression. The correlation between PVT1 expression and ABI, the influencing factors of LEASO through multivariate logistic regression analysis, and a diagnostic prediction for the disease in conjunction with laboratory test indicators were explored. The goal is to offer new insights into LEASO's causes and development and new approaches for its early prevention and treatment.

## PATIENTS AND METHODS

### Clinical data of patients and grouping

A total of 133 LEASO patients at the Qingdao Municipal Hospital from July 2021 to April 2022 were included after screening the inclusion and exclusion criteria. Among them, 44 were complicated with CAD (CAD history detected by coronary artery multi-slice spiral computed tomography imaging or coronary angiography) and defined as the LEASO with CAD group; the remaining 89 patients who have not undergone CAD and other arterial diseases were denoted as the LEASO group. The LEASO group patients were further divided into seven groups according to their Rutherford scores (Table 1): Rutherford grade 0 (nine cases, 10.11%), Rutherford grade 1 (nine cases, 10.11%), Rutherford grade 2 (six cases, 6.74%), Rutherford grade 3 (34 cases, 38.20%), Rutherford grade 4 (19 cases, 21.35%), Rutherford grade 5 (six cases, 6.74%), and Rutherford grade 6 (six cases, 6.74%) according to the Rutherford grading (Table 1); regarding the lesion location, they were divided into unilateral (48 cases) and bilateral (41 cases), and the unilateral affected limb group was further divided into the left (28 cases) and right (20 cases).(The specific grouping is shown in Fig. 1). This study was approved by the Ethics Committee of Qingdao Municipal Hospital. Signed written informed consent was obtained from all participants before the experiment.

Inclusion criteria for the experimental group were: (1) Patients diagnosed with LEASO by lower limb arteriography or Doppler ultrasonography of the lower extremity arteries at the Department of Vascular Surgery of Qingdao Municipal Hospital; (2) those who completed medical records and relevant auxiliary examinations; (3) participants voluntarily joined the study and gave informed consent. Exclusion criteria were: (1) Patients had a history

**Table 1  Rutherford classification reference table.**

| Grade | Category | Clinical description | Objective criteria |
|---|---|---|---|
| 0 | 0 | Asymptomatic–no hemodynamically significant occlusive disease | Normal treadmill or reactive hyperemia test |
|  | 1 | Mild claudication | Completes treadmill exercise[b]; AP after exercise >50 mmHg but at least 20 mmHg lower than resting value |
| I | 2 | Moderate claudication | Between categories 1 and 3 |
|  | 3 | Severe claudication | Cannot complete standard treadmill exercise[b] and AP after exercise <50 mmHg |
| II[a] | 4 | Ischemic rest pain | Resting AP <40 mmHg, flat or barely pulsatile ankle or metatarsal PVR; TP <30 mmHg |
| III[a] | 5 | Minor tissue loss–nonhealing ulcer, focal gangrene with diffuse pedal ischemia | Resting AP <60 mmHg, ankle or metatarsal PVR flat or barely pulsatile; TP <40 mmHg |
|  | 6 | Major tissue loss extending above TM level, functional foot no longer salvagcable | Same as category 5 |

**Notes.**

AP, Ankle pressure; PVR, pulse volume recording; TP, toe pressure; TM, transmetatarsal.

[a] Grades II and III, categories 4, 5, and 6, are embraced by the term chronic critical ischemia.

[b] Five minutes at 2 mph on a 12% incline.

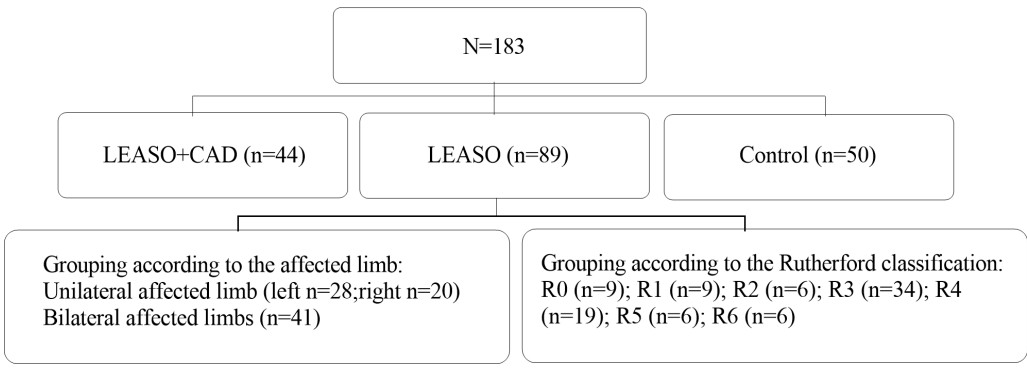

**Figure 1  Experimental grouping flowchart.**

of malignant tumors; (2) individuals diagnosed with arterial diseases besides LEASO and CAD.

Fifty healthy individuals who came to our hospital for physical examination during the same period were randomly included and defined as the Healthy Control group. The group included 25 males and 25 females, with an average age of 72.20 ± 8.09 years. Inclusion criteria were: (1) Participants who were willing to join in the experiment and submitted informed consent; (2) those who were previously healthy; (3) individuals who completed

relevant auxiliary examination information. Exclusion criteria involved: Participants who had basic diseases that may affect the expression of PVT1 in serum (tumors, CAD, *etc.*).

When admission was received from all participants during physical examination, fasting venous blood was drawn to detect the expression of PVT1 by quantitative reverse transcriptase-polymerase chain reaction (qRT-PCR). The brachial artery pressure and ankle artery pressure of all participants were measured with a sphygmomanometer. The clinical data and laboratory test results (including blood tests, liver and kidney function, and blood lipids) of all subjects were collected.

## Detection of relative lncRNA PVT1 expression using qRT-PCR

Fasting venous blood was taken from all included subjects in the morning and centrifuged. The supernatant was collected and stored in liquid nitrogen. RNA was extracted from the samples using TRIzol total RNA extraction reagent. The samples were ground to powder, and an RNA extraction solution was added. The chloroform solution was poured and mixed thoroughly. The mixture was centrifuged, and the upper aqueous phase was transferred to a new centrifuge tube. Isopropanol was added and mixed, then centrifuged. Ethanol liquid was discarded, and the remaining liquid was dissolved in water and stored in a $-80\ °C$ environment. The concentration and purity of RNA were measured using NanoDrop® ND-2000 to check the quality of RNA. The first strand cDNA synthesis kit (Takara PrimeScript RT reagent Kit) was used for reverse transcription. The following reagents were added to a 0.2 ml PCR tube: 1 μL gDNA Eraser, 2 μL 5X gDNA Eraser Buffer, 5 μL of total RNA, and 2 μL of Rnase-free ddH$_2$O. Then the solution was incubated in a water bath at $42\ °C$ for 2 min. After centrifugation, the following reagents were added: 4.0 μL of 5X PrimeScript Buffer 2, 4.0 μL of RT Primer Mix, 1.0 μL of Rnase-free ddH$_2$O, 1.0 μL PrimeScript RT Enzyme Mix I, and the mixture with a total volume of 20.0 μL was obtained. Then it was incubated in a water bath at $37\ °C$ for 15 min, $85\ °C$ for 5 s, and $4\ °C$ for 10 min. The samples were reverse transcribed, and the cDNA samples were diluted 10 times before being tested. Real-time qRT-PCR analysis was performed using Takara TB Green Premix Ex Taq II (2X). An internal reference was introduced to correct differences in cDNA content among samples. The original detection results of RealTime qRT-PCR were calculated according to the relative quantification formula. In our experiment, we utilized GAPDH as an internal reference gene to normalize the expression of our target gene. GAPDH is a widely used reference gene due to its relatively stable expression across various experimental conditions. By measuring the expression level of GAPDH and comparing it to that of our target gene, we were able to effectively normalize the expression of our target gene. This approach helps to reduce experimental bias and improve the overall quality of our data. (The main reagents and instruments, as well as the sequences of PVT1 and primers, are referred to in Tables 2–4. The PCR reaction conditions are shown in Table 5).

## Measurement of ABI

ABI, also known as the Ankle-Brachial Pressure Index, is the blood pressure ratio in the lower leg to that in the upper arm. If the blood pressure in the lower leg is lower than that in the upper arm, it may indicate peripheral vascular disease. The index is calculated as

**Table 2   Main experimental reagents and manufacturer part numbers.**

| Reagent name | Manufacturer | Part number |
|---|---|---|
| Trizol Total RNA Extraction Reagent | Tiangen Biotech (Beijing) Co., Ltd. | |
| PrimeScript™ RT reagent Kit with gDNA Eraser | TaKaRa | RR047A |
| SYBR® Premix Ex Taq™ II (Tli RNaseH Plus) ROX plus | TaKaRa | RR820A |
| Primer synthesis | GENEWIZ | |

**Table 3   Main experimental instruments and model and manufacturer.**

| Instrument name | Instrument model | Manufacturer |
|---|---|---|
| PCR instrument | MP-32 | Hangzhou Mio Instrument Co., Ltd. |
| Mini centrifuge | Mini-10K+C | Hangzhou Mio Instrument Co., Ltd. |
| Nanodrop spectrophotometer | | Thermo Fisher |
| UV analyzer | JY02S | Beijing Junyi Dongfang Electrophoresis Equipment Co., Ltd. |
| Fluorescent quantitative PCR instrument | ABI7500 | Applied Biosystems |
| Sphygmomanometer | YE666AR | Jiangsu Yuyue Medical Equipment Co., Ltd. |

**Table 4   Primers for PVT1 and GAPDH.**

| Primer name | Sequence 5′–3′ |
|---|---|
| PVT1-F | GGTTCCACCAGCGTTATTC |
| PVT1-R | CAACTTCCTTTGGGTCTCC |
| GAPDH-F | GCTCTCTGCTCCTCCCCTGTTC |
| GAPDH-R | ACGACCAAATCCGTTGACTC |

ABI = ankle artery pressure/brachial artery pressure. After resting quietly for 5–10 min, the patient was asked to take the supine position. The blood pressure cuff was wrapped around the upper arm to ensure the cuff was close to the skin and tightened appropriately (generally allowing a finger), and the blood pressure meter was placed properly with the cuff. The lower edge was 2–3 cm above the elbow fossa, and the brachial artery pressure was measured. When measuring ankle artery pressure, the position of the blood pressure meter was coordinative with the cuff's lower edge about 3–4 cm away from the inner ankle, and the tightness of the cuff was appropriate (similar as mentioned).

## Statistical analysis

The data were analyzed using SPSS 23.0 and expressed as mean ± standard deviation. Binary variables were tested by the chi-square test. The PVT1 expression levels in control, LEASO, and LEASO with CAD groups were compared using one-way ANOVA. *T*-test analysis was used to assess the differences in lncRNA PVT1 expression levels and ABI between the single and bilateral affected limb groups and between the left and right lower limb groups within the LEASO group. The relationship between PVT1 expression and ABI

**Table 5  Preparation of PCR reaction mixture.**

| Reaction Component | Concentration | Volume (μl) |
| --- | --- | --- |
| SybrGreen qPCR Master Mix | 2X | 10 |
| ROX | 50X | 0.4 |
| GAPDH-F (10uM) | 10 μM | 0.5 |
| GAPDH-R (10uM) | 10 μM | 0.5 |
| ddH20 | | 4.6 |
| Template (cDNA) | | 4 |
| Total | | 20 μl |

was analyzed using linear correlation. Logistic regression was employed to identify LEASO risk factors, and the ROC curve and area under the curve were used for diagnosis and prediction. $P < 0.05$ was considered statistically significant. Youden Index is a method for evaluating the authenticity of screening tests, and calculated as Youden Index = (sensitivity + specificity) - 1, with an index range of 0–1. The higher index indicates better authenticity of the tests.

# RESULTS

## Baseline characteristics of participants

Patient baseline characteristics were analyzed, including age, gender, smoking status, drinking, hypertension, diabetes, and body mass index. There were no significant differences in age, gender, hypertension, drinking habits, and body mass index among the three groups ($P > 0.05$), indicating good comparability. The LEASO with the CAD group had the highest proportion of patients with diabetes and smoking habits. The LEASO group had significantly more patients with hypertension and diabetes than the control group ($P < 0.05$) (Table 6).

## Expression of PVT1 in serum
### The expression of PVT1 in three groups

QRT-PCR was used to measure lncRNA PVT1 expression levels in serum samples from 183 individuals, including the healthy controls and LEASO patients. The Shapiro–Wilk method was adopted to test the normal distribution of PVT1 expression in the three groups. It was shown that only the PVT1 expression in the LEASO group followed a normal distribution ($P>0.05$). The non-normally distributed PVT1 expression in the three groups was subjected to non-parametric testing. PVT1 expression differed significantly among healthy control, LEASO and LEASO with CAD groups. The highest expression was in the LEASO with the CAD group, followed by the LEASO group and then the control group ($P < 0.05$). This trend was exhibited in a scatter plot (Table 7, Fig. 2).

### The expression of PVT1 between the left and right affected limbs in the LEASO group

The Shapiro–Wilk method was applied to test the normal distribution of PVT1 expression in the left and right affected limb groups and found that PVT1 expression was normally

**Table 6  Baseline data table.**

| Variables | Control (n = 50) | LEASO (n = 89) | LEASO and CAD (n = 44) | $c^2$/F | P |
|---|---|---|---|---|---|
| Age (year) | 72.20 ± 8.09 | 72.47 ± 9.33 | 74.55 ± 9.80 | 0.961 | 0.385 |
| Gender(Male/Female) | 25/25 | 57/32 | 23/21 | 3.199 | 0.202 |
| Hypertension n(%) | 19(38.0%) | 57(64.0%) | 33(75.0%) | 14.748 | 0.001 |
| Wine n(%) | 10(20.0%) | 21(23.6%) | 8(18.2%) | 0.585 | 0.746 |
| Smoker n(%) | 17(34.0%) | 31(34.8%) | 14(31.8%) | 0.12 | 0.942 |
| Diabetes mellitus n(%) | 11(22.0%) | 42(47.2%) | 24(54.5%) | 12.031 | 0.002 |
| BMI (kg/m$^2$) | 24.49 ± 3.17 | 24.16 ± 3.45 | 24.13 ± 2.84 | 0.207 | 0.813 |

**Table 7  Normal distribution test of lncRNA PVT1 expression in three groups.**

| Group | Number of cases | Rank average |
|---|---|---|
| Control | 50 | 34.52 |
| LEASO | 89 | 96.36 |
| LEASO with CAD | 44 | 148.5 |
| Total | 183 | |

distributed ($P > 0.05$) in the two groups. In addition, no significant difference was detected by the $t$-test ($P > 0.05$) (Table 8, Fig. 3).

### *The expression of PVT1 between the unilateral and nilateral affected limbs in the LEASO group*

Similarly, the PVT1 expression in single and double limb groups displayed normal distribution by the Shapiro–Wilk method and no significant difference according to $t$-text ($P > 0.05$) (Table 9, Fig. 4).

### The relationship among Rutherford classification, ABI and PVT1 expression

LEASO patients were categorized into seven groups based on the Rutherford classification (*Rutherford et al., 1997*). PVT1 expression and ABI values were measured in each group. PVT1 expression increased with Rutherford grade and was negatively correlated with ABI ($r = -0.710$, $P < 0.01$) (Table 10, Figs. 5–6).

### Analysis of influencing factors of LEASO

Univariate logistic regression analysis showed that triglycerides (TG), cholesterol (TC), high-density lipoprotein cholesterol (HDL-C), low-density lipoprotein cholesterol (LDL-C), C-reactive protein (CRP), and PVT1 were significant factors for LEASO ($P < 0.001$) in the healthy controls and LEASO patients. The laboratory indicators with statistically significant differences in the univariate binary logistic regression analysis were selected for the multivariate binary logistic regression. The results demonstrated no statistical
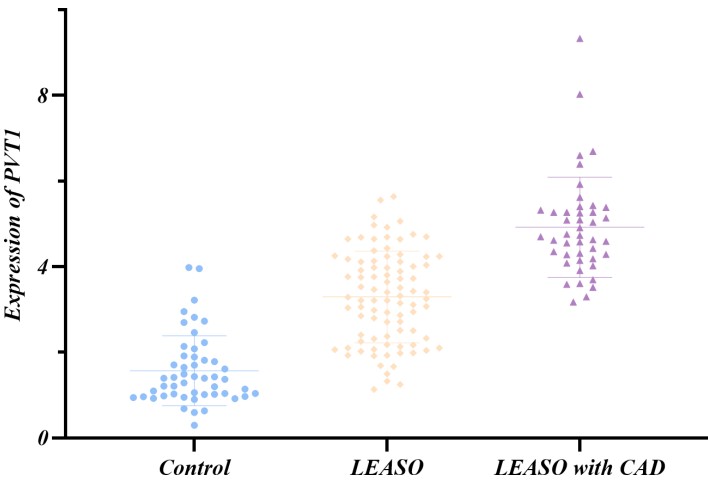

**Figure 2** Relative expression of PVT1 in serum of healthy control group, LEASO group, and LEASO with CAD group.

**Table 8  Relative expression of PVT1 in left and right affected limbs in the LEASO group.**

| Group | Number of cases | PVT1 |
|---|---|---|
| Left Lower Limb group | 28 | $3.06 \pm 1.17$ |
| Right Lower Limb group | 20 | $3.23 \pm 1.06$ |
| *T* value | | $-0.519$ |
| *P* value | | 0.606 |

significance in indicators ($P > 0.05$). The logistic regression analysis method was changed to forward stepwise (likelihood ratio) binary logistic regression. According to the Omnibus test of model coefficients, the $-2$ log-likelihood in model 4 was 50.439, lower than that in models 1–3; the *R*-squared was 0.838, the highest among all models. Thus, model 4 was selected for multivariate binary logistic regression. TG (OR $=2.972$, 95%CI [1.159–7.618]), LDL-C (OR $=6.655$, 95%CI [1.490–29.723]), CRP (OR $=1.686$, 95%CI [1.218–2.335]), and PVT1 (OR $=2.885$, 95%CI [1.350–6.167]) were independent risk factors for LEASO (Table 11, Fig. 7).

## Comparison of the efficacy of PVT1 combined with laboratory indicators in predicting LEASO

A ROC curve was drawn using significant indicators from the multivariate logistic regression. The area under the curve for predicting LEASO was 0.902 for PVT1 alone and 0.976 for PVT1+TG+LDL-C+CRP (95%CI [0.951–1.000], $P < 0.001$) (Table 12, Fig. 8).
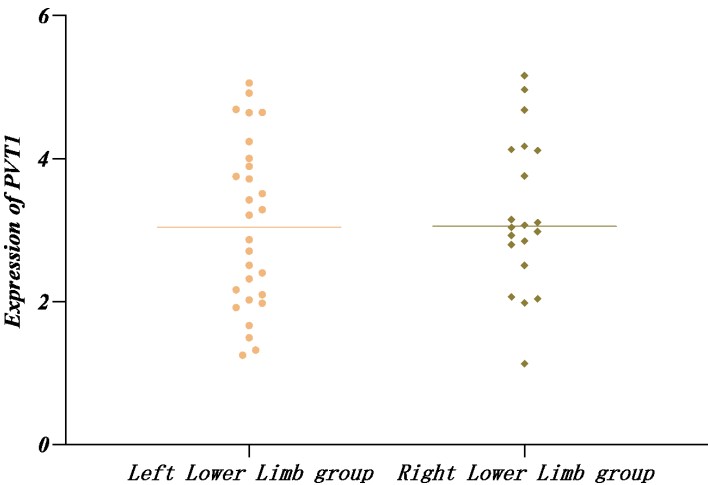

**Figure 3** Relative expression of PVT1 in left and right affected limbs in the LEASO group.

**Table 9** Relative expression of PVT1 in unilateral and bilateral affected limbs in the LEASO group.

| Group | Number of cases | PVT1 |
|---|---|---|
| Unilateral Lower Limb group | 48 | $3.13 \pm 1.12$ |
| Bilateral Lower Limb group | 41 | $3.48 \pm 1.00$ |
| *T* value | | $-1.516$ |
| *P* value | | 0.133 |

## DISCUSSION

LEASO is a common vascular disease that can cause disability and death. Its incidence rates increase with age (*Takahara, 2021*; *Frank et al., 2019*). In China, about 45.3 million patients conformed to LEASO (*Wang et al., 2019*). Even in developed countries, many LEASO patients require amputation due to delayed diagnosis and treatment (*Nickinson et al., 2020*). In this sense, standardized treatment should be carried out as early as possible for LEASO patients. In this article, no statistically significant difference was found in people with hypertension between the healthy control group and the groups with LEASO and LEASO with CAD. Bias may exist in the results due to the small sample size. The proportion of patients with smoking behavior and diabetes in the two groups was significantly higher than in the control group.

Given that lncRNAs are further conformed to play a role in cardiovascular diseases, such as AS, they are increasingly used as a tool or target for diagnosing and treating vascular surgery. *Sun et al. (2020)* proposed that PVT1 promotes vascular endothelial cell proliferation by inhibiting miR-190a-5p, contributing to chronic heart failure. Researchers found increased PVT1 expression in peripheral blood monocytes of CAD patients (*Nowrouzi-Sohrabi et al., 2022*). *Lu et al. (2021)* reported that CASC8, CASC11 and PVT1 genetic polymorphisms were associated with the Gensini score in CAD patients.

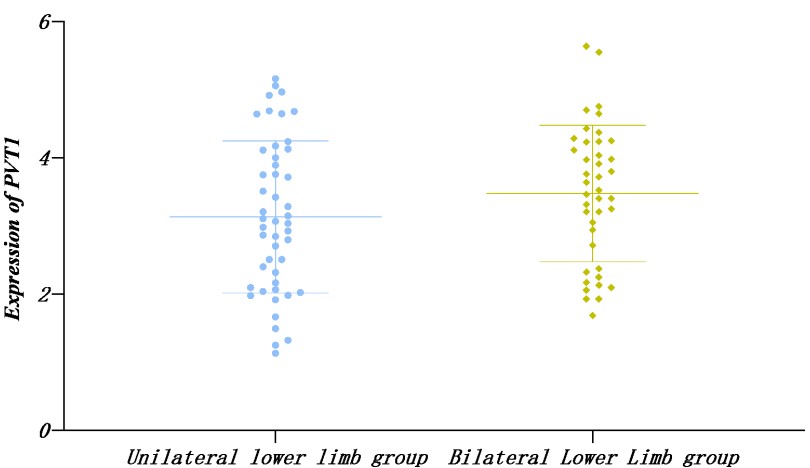

**Figure 4** Relative expression of PVT1 in unilateral and bilateral affected limbs in the LEASO group.

**Table 10  Relationship between Rutherford classification and ABI, PVT1 expression.**

| Rutherford Stages | 0 | 1 | 2 | 3 | 4 | 5 | 6 |
|---|---|---|---|---|---|---|---|
| Number of cases (%) | 9(10.11%) | 9(10.11%) | 6(6.74%) | 34(38.20%) | 19(21.35%) | 6(6.74%) | 6(6.74%) |
| ABI | $0.86 \pm 0.05$ | $0.66 \pm 0.07$ | $0.61 \pm 0.07$ | $0.55 \pm 0.08$ | $0.42 \pm 0.09$ | $0.30 \pm 0.06$ | $0.08 \pm 0.03$ |
| PVT1 | $1.80 \pm 0.46$ | $2.79 \pm 0.71$ | $2.90 \pm 0.68$ | $3.11 \pm 0.92$ | $3.87 \pm 0.80$ | $4.19 \pm 0.28$ | $4.96 \pm 0.47$ |

The Gensini score is a commonly used method for quantifying angiographic AS, and a zero value indicates the absence of atherosclerotic disease. The score considers the degree and location of artery narrowing, thus, is regarded as an effective tool for assessing the severity of CAD.

Research showed that PVT1 was highly expressed in the serum of CAD patients and was positively correlated with the Gensini score, suggesting that PVT1 may be important in AS progression. Examining PVT1 expression levels in the serum can help differentiate between mild and severe CAD (*Quan et al., 2020*). Our findings align with previous research. It was found that PVT1 expression was the highest in the LEASO combined CAD group ($P < 0.01$) and higher in the LEASO group than in the healthy control group ($P < 0.01$).

In the LEASO group, the expression of PVT1 gradually increased with the Rutherford scores. It was speculated that a certain relationship might exist between PVT1 and the severity of LEASO lesions. No significant difference was observed in the PVT1 expression levels in the two comparative groups within the LEASO group ($P > 0.05$). This observation suggests that PVT1 expression may be related to the extent and severity of AS lesions rather than their location.

The current guidelines indicate that ABI is the primary non-invasive diagnostic tool for the initial diagnosis of LEASO (*Aboyans et al., 2018*). The index is typically measured by comparing the patients' ankle artery pressure to their brachial pressure. A result below 0.9 is

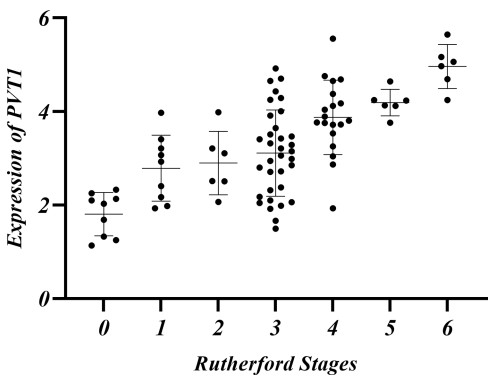

**Figure 5** Relative expression of PVT1 in Rutherford classification in the LEASO group.

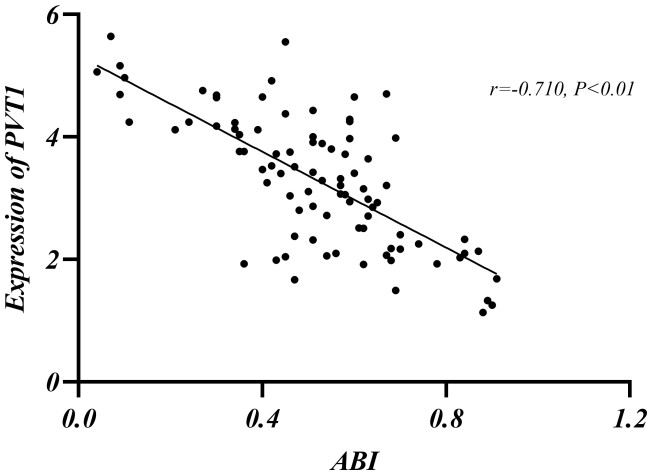

**Figure 6** Relationship between ABI and PVT1 expression.

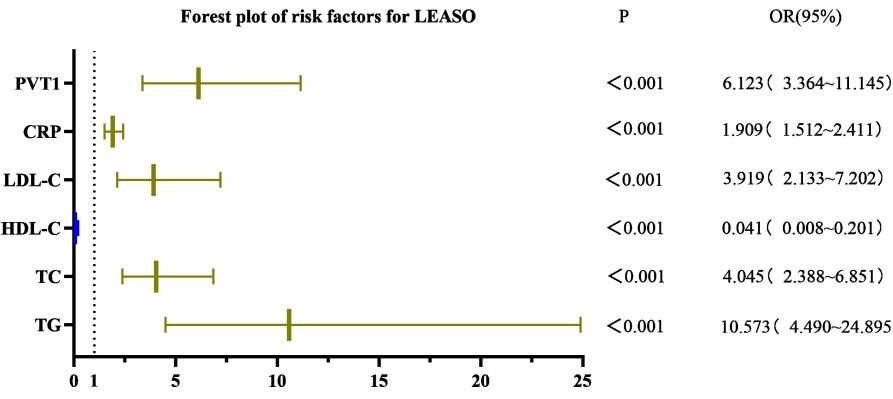

**Figure 7** Forest plot of univariate logistic regression analysis of factors affecting LEASO.

**Table 11  Univariate and multivariate logistic regression analysis of factors affecting LEASO.**

| Variables | Univariate analysis | | Multivariate analysis | |
|---|---|---|---|---|
| | OR(95% CI) | *p* value | OR(95% CI) | *p* value |
| TG | 10.573(4.490–24.895) | <0.001 | 2.972(1.159–7.618) | 0.023 |
| TC | 4.045(2.388–6.851) | <0.001 | (–) | |
| HDL-C | 0.041(0.008–0.201) | <0.001 | (–) | |
| LDL-C | 3.919(2.133–7.202) | <0.001 | 6.655(1.490–29.723) | 0.013 |
| BUN | 1.086(0.868–1.359) | 0.470 | (–) | |
| CR | 1.000(0.994–1.005) | 0.907 | (–) | |
| UA | 1.003(0.999–1.008) | 0.127 | (–) | |
| Neutrophil count | 0.890(0.756–1.047) | 0.159 | (–) | |
| Lymphocyte count | 0.758(0.359–1.599) | 0.467 | (–) | |
| Platelet count | 0.997(0.993–1.001) | 0.202 | (–) | |
| CRP | 1.909(1.512–2.411) | <0.001 | 1.686(1.218–2.335) | 0.002 |
| PVT1 | 6.123(3.364–11.145) | <0.001 | 2.885(1.350–6.167) | 0.006 |
| NLR | 0.832(0.613–1.130) | 0.239 | (–) | |
| PLR | 0.997(0.990–1.004) | 0.400 | (–) | |

**Table 12  Comparison of the efficacy of various laboratory indicators in predicting LEASO.**

| Variables | AUC | 95% CI | cut-off value | *p* value | Sensitivity (%) | Specificity (%) | Jordan index |
|---|---|---|---|---|---|---|---|
| TG | 0.915 | 0.863–0.967 | 1.855 | <0.001 | 84.30 | 90.00 | 0.743 |
| LDL-C | 0.730 | 0.646–0.814 | 2.535 | <0.001 | 76.40 | 60.00 | 0.364 |
| CRP | 0.909 | 0.861–0.956 | 4.970 | <0.001 | 78.70 | 96.00 | 0.747 |
| PVT1 | 0.902 | 0.848–0.956 | 1.916 | <0.001 | 93.30 | 78.00 | 0.713 |

abnormal, and below 0.4 indicates severe ischemia (*Frank et al., 2019*). In this experiment, the Rutherford classification was adopted to grade patients in the LEASO group (*Rutherford et al., 1997*). A negative correlation was found between ABI values and PVT1 expression levels (Table 10, Figs. 5–6). The lower ABI values indicate more severe disease associated with the higher PVT1 expression. The high PVT1 expression was seen in the serum samples of patients with LEASO. Additionally, a certain correlation was observed between PVT1 expression and Rutherford classification, and ABI values were negatively correlated with PVT1 expression. This study is the first to demonstrate a negative correlation between PVT1 and ABI.

PVT1 and laboratory indicators were jointly analyzed by logistic regression. The results of univariate binary logistic regression analysis showed that TG (OR =10.573, 95%CI [4.490–24.895]), TC (OR =4.045, 95%CI [2.388–6.851]), HDL-C (OR =0.041, 95%CI [0.008–0.201]), LDL-C (OR =3.919, 95%CI [2.133–7.202]), CRP (OR =1.909, 95%CI [1.512–2.411]), and PVT1 (OR =6.123, 95%CI [3.364–11.145]) were factors affecting LEASO. Multivariate binary logistic regression analysis was conducted based on the significant indicators in the univariate analysis. It was revealed that TG (OR =2.972, 95%CI [1.159–7.618]), LDL-C (OR =6.655, 95%CI [1.490–29.723]), CRP (OR =1.686,

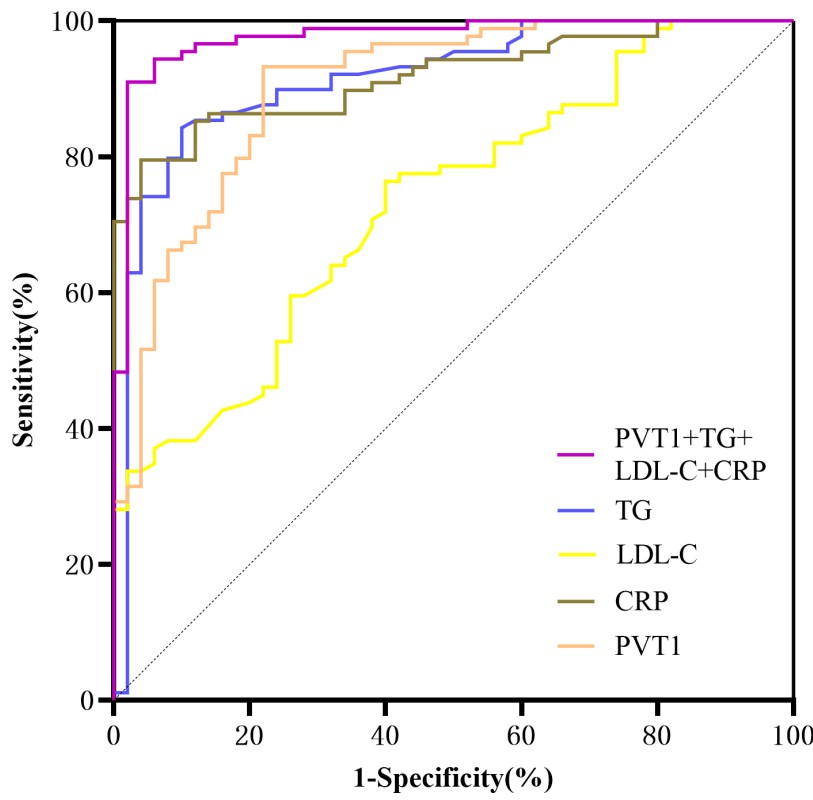

**Figure 8** ROC curves for prediction of LEASO by various laboratory indicators.

95%CI [1.218–2.335]), and PVT1 (OR =2.885, 95%CI [1.350–6.167]) were independent risk factors for the occurrence of LEASO (Table 11, Fig. 7). Previous research has shown that abnormal blood lipids and inflammation play a key role in the development of AS (*Kong et al., 2022*). Abnormal blood lipids are manifested as the chronic accumulation of lipid-rich plaques in the arteries (*Begum et al., 2019*; *Wen et al., 2019*), including unhealthy levels of one or more lipids in the blood circulation, such as high levels (exceeding normal values) of TG, TC, LDL-C, and HDL-C. Abnormal blood lipids and lipid metabolism disorders are major causes of AS development (*Pol et al., 2018*). Epidemiological studies have linked high LDL-C levels to myocardial infarction and ischemic stroke (*Sharrett et al., 2001*; *Mortensen & Nordestgaard, 2020*), and LDL-C is a major risk factor for cardiovascular events (*Ference et al., 2017*; *Imes & Austin, 2013*). Increasing HDL-C levels may help prevent AS (*Catapano et al., 2016*; *Mach et al., 2020*), which was supported by our results. The univariate logistic regression analysis unveiled that LDL-C was an independent risk factor for LEASO while HDL-C was a protective factor. TG can increase AS risk by impairing microcirculation and enhancing AS lipoprotein-endothelium interactions (*Rosenson et al., 2001*). One study found that higher TG levels were associated with a 72% increased risk of coronary heart disease (*Sarwar et al., 2007*). This experiment found that TG was an independent risk factor for LEASO. In addition, inflammation acts as another important driver of cardiovascular risk. CRP is one of the most important inflammatory markers and has been reported to be

associated with cardiovascular events (*Arima et al., 2008*; *Liu et al., 2020*). LncRNA PVT1 is also an independent risk factor for LEASO. PVT1 combined with laboratory indicators to predict LEASO has a large area under the ROC curve. The prediction results show that the combined method is more sensitive than a single indicator. PVT1 can be used as a diagnostic indicator for LEASO, with a Youden's index of 0.713 for predicting the disease (Table 12, Fig. 8).

This study is innovative in that it is the first to show elevated lncRNA PVT1 expression in LEASO patients' serum, with the highest levels in LEASO patients with CAD. Combing PVT1 with laboratory indicators has strong sensitivity for diagnosing and predicting LEASO. This study has limitations. Primarily, bias may be introduced due to a small sample size from a single hospital; Only high lncRNA PVT1 expression in LEASO patients' serum was detected, and its specific role in AS pathogenesis was neglected. Further research is needed to understand the regulatory mechanisms of lncRNA PVT1 in AS.

## CONCLUSION

In summary, lncRNA PVT1 may play a role in LEASO development and is associated with the degree of AS. It is highly expressed in LEASO patients and is an independent risk factor for the disease. PVT1 can be combined with effective laboratory indicators as a marker for diagnosing LEASO. However, the specific regulatory role of PVT1 in the pathogenesis of AS still needs further experimental research.

## ACKNOWLEDGEMENTS

We would like to thank all the research staff who made it possible to perform this study.

### Funding

This study was supported by the National Natural Science Foundation of China (Grant 81871187). The funders had no role in study design, data collection and analysis, decision to publish, or preparation of the manuscript.

### Grant Disclosures

The following grant information was disclosed by the authors:
National Natural Science Foundation of China: 81871187.

### Competing Interests

Xiaoqing Yuan is employed by Guangzhou Kingmed Diagnostics Group. Other authors declare that there are no competing interests.

### Author Contributions

- Xiaoxue Su conceived and designed the experiments, performed the experiments, prepared figures and/or tables, and approved the final draft.

- Xiaoqing Yuan conceived and designed the experiments, prepared figures and/or tables, and approved the final draft.
- Fenghui Li performed the experiments, prepared figures and/or tables, and approved the final draft.
- Guinan Yang performed the experiments, prepared figures and/or tables, and approved the final draft.
- Liunianbo Du analyzed the data, prepared figures and/or tables, and approved the final draft.
- Fule Zhao analyzed the data, prepared figures and/or tables, and approved the final draft.
- Rui Zhao analyzed the data, prepared figures and/or tables, and approved the final draft.
- Minghui Ou conceived and designed the experiments, authored or reviewed drafts of the article, and approved the final draft.

## Human Ethics

The following information was supplied relating to ethical approvals (i.e., approving body and any reference numbers):

Approval Document 2022 Linshen Zi No. 041 from the Ethics Committee of Qingdao Municipal Hospital

## Data Availability

The original measurement values are available in the Supplementary Files.

## Supplemental Information

Supplemental information for this article can be found online at http://dx.doi.org/10.7717/peerj.16057#supplemental-information.

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
