# Peer review of "Expression level and clinical significance of LncRNA PVT1 in the serum of patients with LEASO"

_PeerJ, doi:10.7717/peerj.16057_

## Round 0.1 · original submission · Major Revisions

Please consider all reviewers' remarks

Reviewer 1 ·

Basic reporting

The paper from Su et al, entitled « Expression level and significance of LncRNA PVT1 in the serum of patients with lower extremity arteriosclerosis obliterans » is interesting, but it lacks different things in the Patients and methods part and in the Result part.

I did not find the legends of the figures.

In the text, in the Result part, the Figures and Tables are not indicated.

Table 1 is indicated lane 102, but it seems to be a mistake because Table 1 is not about the Rutherford grade.

Experimental design

• 2. Experimental design
The authors don’t say how they calculate the relative expression of PVT1. And on Figures, there is no unit for the relative expression of PVT1, is it AU (Arbitrary Unit)? Sometimes it is written, Expression of PVT1 or just PVT1.

I did not find the legends of the figures.

Validity of the findings

The RNA extraction was made by Trizol, did the authors check the quality of the RNA by instance with the Bioanalyser from Agilent, in order to be sure that they are not degraded.
The authors don’t indicate lane 30 which kit and conditions they use for the reverse transcription of total RNA.

Additional comments

Lane 129 : it should be Trizol and not TRNzol. Could the authors change it please ?
Lane 205 : « Lnc RNA help maintain this balance. » should be change in : « Lnc RNA help to maintain this balance. »
Lane 219 « AS » is not define, is it Aortic Stenosis ?
Could the authors explain by a sentence the Gensini score and the Jordan index, please ?

Reviewer 2 ·

Basic reporting

I am pleased to serve as a reviewer for your original article entitled "Expression level and significance of LncRNA PVT1 in the serum of patients with lower extremity arteriosclerosis obliterans”.

Your study addresses a crucial gap in the current literature by examining the expression profile of a specific lncRNA in patients with LEASO and the comparison of the efficacy of PVT1 combined with laboratory indicators in predicting LEASO.
As a reviewer, my primary goal is to provide constructive feedback to enhance the scientific rigor and impact of your study. In the subsequent sections of this review, I will critically give a basic reporting, evaluate the experimental design and the validity of the findings. I will also offer suggestions for improving the clarity and presentation of your work, with the aim of enhancing its scientific value.

Authors should carefully revise the text for improved clarity. Your manuscript requires improvement to ensure clear understanding by an international audience. There are several instances where the language could be enhanced for better comprehension. For example, lines 136-138 contain phrasing that makes it difficult to understand the intended meaning. Additionally, in line 142, please provide a sentence explaining the calculation of the ABI. For instance, in the "Patient and Methods" section, there are very long sentences enclosed in brackets, which should be revised for better readability.
Repetitions in the text, such as presenting results in the discussion section, should be avoided. Additionally, all abbreviations must be defined when they first appear in the text, rather than only in the discussion section. I recommend having a colleague who is proficient in English and familiar with the subject matter review your manuscript, or consider engaging a professional editing service.

To facilitate reading, consider adding asterisks (*) to indicate the significance of figures, for example. Moreover, it is important to integrate the figures into the results section to provide a clearer flow of information for readers.
Lastly, please ensure that the figures meet the required resolution standards for clear visibility in the article.
By addressing these language and presentation issues, the manuscript will become more accessible, engaging, and aligned with professional publishing standards.

Experimental design

While the subject matter of the article aligns with the general aims and scope of PeerJ, there are certain areas that require improvement to meet the specific standards and requirements of the journal.

I would like to commend you for undertaking a study that investigates the expression of a specific long non-coding RNA, the lncRNA PVT1, in individuals affected by a lower extremity arteriosclerosis obliterans. This research area holds great promise for advancing our understanding of the molecular mechanisms underlying this complex vascular disease.
Atherosclerosis remains a leading cause of mortality worldwide, posing a significant burden on public health systems. While extensive research has elucidated the role of protein-coding genes and protein regulators in atherosclerosis, the contribution of non-coding RNAs, particularly lncRNAs, is gaining increasing attention. Identifying serum markers to detect LEASO is essential.

Validity of the findings

Have you identified potential confounders and adjusted for them in your statistical models? If so, could you please provide details on the methods used for identification and adjustment?

·

Basic reporting

The manuscript by Su et al., entitled. "Expression level and significance of LncRNA PVT1 in the serum of patients with lower extremity arteriosclerosis obliterans” is well written and gives new highlight of the importance of lncRNA in human physiology. The objective of this scientific paper is to understand the role of lncRNA PVT1 in LEASO and to investigate its clinical significance. The paper has a clear and concise research objective, which is a key strength. The patient grouping, methodology, and results were well structured and presented clearly.
Still, although of putative interest for the readers of peerJ, the present study requires some modifications in the methods and data normalization before publication.

Minor points:
1. At times, the discussion seems to drift towards providing excessive background information. While it is necessary to include some background, it would be helpful to more directly tie this information back to the specific results of the current study. In example, line 214, location and lebght of PVT1 should be in the introduction, like pathogenesis of tumors line 215.
2. The authors could emphasize more the novelty of their study in the discussion - that they are the first to demonstrate elevated lncRNA PVT1 expression in LEASO patients’ serum. This is currently mentioned only in the last paragraph and could be brought up earlier to highlight the significance of the study.
3. The authors should consider discussing any potential applications of their work. For example, how might their findings be used in future diagnostic or treatment protocols?
4. The figures need to be similar (sometimes it is just PVTI, sometimes it is expression of PVT1). You should also add more details in the figure legends, it will be easier for the reader. In example, figure 5 you could add the ABI signification from material and methods.

Experimental design

Major concern:

In your manuscript, I noticed that you have utilized U6 as a reference gene to normalize PVT1 expression in serum samples. However, prior research by Wang et al. has suggested that U6 could be downregulated in cardiovascular diseases, such as heart failure and hypertension. As such, this downregulation could possibly induce an artificial upregulation of PVT1 in patient samples (https://www.ncbi.nlm.nih.gov/pmc/articles/PMC6006091/).
Did you consider using an alternate reference gene for normalization in patient samples (or more than one)? I noticed from reference 10 of your paper (Quan W et al.) that they normalized their data using GAPDH. Could you explain the reasoning behind your choice of U6 over GAPDH or other potential reference genes?
Moreover, Heidi Schwarzenbach advocates for normalization with reference genes belonging to the same RNA class. Considering U6 is a small RNA of approximately 106 nucleotides in length, it seems to be potentially mismatched as a reference gene for normalizing PVT1, a long non-coding RNA that spans 1716 nc (https://www.ncbi.nlm.nih.gov/pmc/articles/PMC4890630/).
Would it be possible for you to provide the raw cycle threshold (ct) values for U6 and PVT1? Additionally, could you present a graph displaying the raw data (maybe expressed with 2^(ct) for U6 expression across various stages, prior to normalization? These additions could significantly enhance the transparency and comprehensibility of your data.

Validity of the findings

no comment

---

## Round 0.2 · Minor Revisions

Dear Dr. Su,

Thank you for your resubmission. I have now received the report from the reviewers. Basically, the revision is now acceptable for publication, but before final acceptance is given, I would appreciate it if you would address the remaining minor issues raised by Reviewer 3 (normalization issue).

I hope that you will be prepared to make the necessary amendments and submit a revised manuscript accompanied by a statement of how you have responded to the Reviewer’s comments. Please copy and paste each and every reviewer's comment above your response. You are also kindly requested to provide a complete tracked changes version of the manuscript in order to make it easier to verify that the required changes have been made.

If you are willing to do this, it would not be necessary for me to return the manuscript to the reviewers, but it could then be accepted for publication.

I look forward to receiving your revision.

Sincerely,
Stefano Menini

Reviewer 2 ·

Basic reporting

I am pleased to serve as a reviewer for your original article entitled "Expression level and significance of LncRNA PVT1 in the serum of patients with lower extremity arteriosclerosis obliterans”.

Your study addresses a crucial gap in the current literature by examining the expression profile of a specific lncRNA in patients with LEASO and the comparison of the efficacy of PVT1 combined with laboratory indicators in predicting LEASO.

The authors have taken into account suggestions for improving the quality and comprehensibility of the manuscript

Experimental design

This research area holds great promise for advancing our understanding of the molecular mechanisms underlying this complex vascular disease.

Atherosclerosis remains a leading cause of mortality worldwide, posing a significant burden on public health systems. While extensive research has elucidated the role of protein-coding genes and protein regulators in atherosclerosis, the contribution of non-coding RNAs, particularly lncRNAs, is gaining increasing attention. Identifying serum markers to detect LEASO is essential.

The authors have taken into account suggestions for improving the quality and comprehensibility of the manuscript

Validity of the findings

The authors have included an explanation of potential confounding factors in the results section, detailing the methods used for identification.

·

Basic reporting

You addressed all my points, and I accept the manuscript for publication

In the qPCR materials and methods section, could you slightly enhance the normalization part? Please specify that you normalized with GAPDH; I didn't notice it in the updated version.

I also quickly created a graph of GAPDH and PVT1 expression using your raw data (I've attached it). GAPDH expression also increased in your Group C. If there's an overall increase in RNA circulation, it might be also interesting. I've seen other studies with a similar pattern, but I haven't encountered it in paper discussions yet. You don't have to add anything from this to the article, but it could be interesting for the next research. It could also be good to have more than one internal control, GAPDH is not enough it is always good to have more than 1 reference gene.

Experimental design

no comment

Validity of the findings

no comment

Additional comments

no comment

---

## Round 0.3 · accepted · Accept

Dear Dr. Su,

Thank you for submitting a revised version of your manuscript. I am pleased to inform you that your manuscript is accepted for publication in PeerJ in its current form.

I thank all reviewers for their effort in improving the manuscript and the authors for their cooperation throughout the review process.

Sincerely yours,
Stefano Menini